# Are Real-World Prosociality Programs Associated with Greater Psychological Well-Being in Primary School-Aged Children?

**DOI:** 10.3390/ijerph20054403

**Published:** 2023-03-01

**Authors:** Jason D. E. Proulx, Julia W. Van de Vondervoort, J. Kiley Hamlin, John F. Helliwell, Lara B. Aknin

**Affiliations:** 1Department of Psychology, Simon Fraser University, Burnaby, BC V5A 1S6, Canada; 2Department of Philosophy & Psychology, University of Waterloo, Waterloo, ON N2L 3G1, Canada; 3Department of Psychology, University of British Columbia, Vancouver, BC V6T 1Z4, Canada; 4Vancouver School of Economics, University of British Columbia, Vancouver, BC V6T 1L4, Canada

**Keywords:** prosocial behavior, well-being, prosociality interventions, prosocial

## Abstract

Quality education can build a sustainable, happier world, but what experiences support student well-being? Numerous laboratory studies suggest that prosocial behavior predicts greater psychological well-being. However, relatively little work has examined whether real-world prosociality programs are associated with greater well-being in primary school-aged children (aged 5–12). In Study 1, we surveyed 24/25 students who completed their 6th Grade curriculum in a long-term care home alongside residents called “Elders,” which offered numerous opportunities for planned and spontaneous helping. We found that the meaning that students derived from their prosocial interactions with the Elders was strongly associated with greater psychological well-being. In Study 2, we conducted a pre-registered field experiment with 238 primary school-aged children randomly assigned to package essential items for children who experience homelessness and/or poverty who were either demographically similar or dissimilar in age and/or gender to them as part of a classroom outing. Children self-reported their happiness both pre- and post-intervention. While happiness increased from pre- to post-intervention, this change did not differ for children who helped a similar or dissimilar recipient. These studies offer real-world evidence consistent with the possibility that engaging in prosocial classroom activities—over an afternoon or year—is associated with greater psychological well-being in primary school-aged children.

## 1. Introduction

Quality education is a central pillar of a well-functioning, sustainable society as it can both uphold social and economic welfare and support children’s psychological well-being [1,2]. Yet, many schools focus on teaching academic subjects (e.g., math, science [3]) and rarely prioritize cultivating social-emotional skills which help children promote and sustain their well-being across their lifetime [1]. As rates of loneliness rise and happiness fall among children [4], how can we re-envision education to promote children’s well-being and positive psychological functioning? One meaningful solution to help children thrive may be to help them support others in their community [5].

A growing body of research suggests that people derive psychological benefits from prosocial action—acts performed with the intention to help others [6]. Indeed, numerous laboratory studies have shown that engaging in prosocial behavior is perceived as meaningful (e.g., [7]) and can boost subjective well-being and positive psychological functioning, both among adults and even toddlers under two years old (e.g., [8,9,10,11,12]). However, are these associations detectable in primary school-aged children (aged 5–12 years) outside the confines of controlled lab studies and in realistic community settings and schools? Using both a correlational study and pre-registered experiment, we tested whether two school programs designed to foster prosocial behavior in children—one completed in an afternoon and the other over a school year—are associated with greater subjective well-being and positive psychological functioning in children.

### 1.1. Prosociality and Well-Being

A number of laboratory studies conducted with adults over the past two decades suggest that engaging in prosociality is a path to greater well-being [10,13]. For instance, a recent registered replication report examined the emotional benefits of prosocial spending—spending money on others as opposed to oneself—in a series of laboratory and online experiments [11]. In the lab experiment, 712 participants were randomly assigned to purchase a goody bag filled with juice or chocolate for either themselves or a sick child at a children’s hospital. Afterward, participants reported their momentary emotions on two validated measures of well-being. Consistent with the conclusion that kindness promotes well-being for the helper, participants randomly assigned to purchase treats for others reported higher levels of happiness. More broadly, a recent meta-analysis examined responses from 16,425 people across 55 studies in which participants were randomly assigned to engage in a prosocial act or not [14]. The authors reported evidence for a small-to-medium effect size when comparing various prosocial acts to a control condition (ref. [14]; see also [12]). Thus, giving to others may reliably boost well-being for adult helpers.

The emotional rewards of prosocial action have also been detected among young children in lab studies. In one lab experiment [8], 20 toddlers under the age of two were introduced to a puppet and, over the span of several minutes, were invited to (a) greet the puppet, (b) receive eight edible treats, (c) watch an experimenter give a treat to the puppet, (d) give a treat provided by the experimenter to the puppet, and (e) give one of their own treats to the puppet. The toddlers’ behavior was video recorded throughout, and facial expressions were later coded for happiness by a team of trained coders. Analyses revealed that toddlers smiled more when giving treats away to the puppet than when receiving treats themselves, both when toddlers gave their own treat or an identical treat provided by the experimenter, suggesting that toddlers find giving emotionally rewarding (ref. [8]; see also [15]). Other research converges with the notion that young children find prosocial action rewarding. For example, two-year-old children in a lab who observe a target in need consistently show signs of increased sympathetic arousal as measured via pupil dilation [16]. Interestingly, this arousal is attenuated when either the child or another helper assists the needy target, suggestive that young children (1) are concerned about others’ well-being, (2) experience relief when others’ needs are met, and (3) are equally relieved whether they or someone else provides the assistance [17].

Despite evidence suggesting that prosocial action can lead to well-being for young children, relatively few studies have examined the associations between prosociality and subjective well-being in primary school-aged children aged 5–12 [18,19,20,21,22,23,24,25,26]. Indeed, a substantial portion of past research has been conducted either with adults or very young children. Moreover, the finding that prosocial action leads to emotional rewards has more often than not been generalized from highly controlled, contrived laboratory studies rather than work conducted in *real-world* community or school settings (however, see [18,19,22,25]). To be sure, controlled experiments allow researchers to probe the causal relationship between prosocial action and happiness with rigorous and precise control. However, these gains in internal validity often come at the expense of external validity and do not account for the messiness of testing and applying these relationships in the wild [27,28]. Thus, findings from past lab-based experiments should not be assumed to generalize to school and community settings without direct testing.

Examining the link between prosociality and well-being in community and school-based settings is critical in light of the various school- and community-based interventions that aim to nurture a prosocial identity in children, often with the expectation that these programs will enhance children’s psychological well-being [29,30,31,32,33]. A wide range of children-focused, school-based prosociality programs have been developed, such as Social Emotional Learning programs, e.g., [32], Intergenerational programs (e.g., [34]), Service Learning and Community Engagement programs (e.g., [35]), Experiential Philanthropy programs (e.g., [36,37], Prosocial Mentorship programs (e.g., [38]), Character Education programs (e.g., [39]), and Moral Learning programs (e.g., [40]). All prosociality-focused interventions rest on the assumption that school-aged children can develop greater prosocial tendencies by learning about or practicing prosocial behavior in some form or another and that these experiences can foster greater well-being. Critically, however, investigations of these programs are limited in several ways.

First, most explorations into the potential benefits of prosociality programs investigate success on alternative markers of well-being, such as lower incidence of behavioral issues and mental health challenges (e.g., symptoms of depression, anxiety [32,41,42]). While better grades, fewer behavioral issues, as well as reduced stress and depression are valuable outcomes, well-being is more than the absence of negative states and challenges; well-being involves experiencing positive emotions and psychological outcomes, such as optimism, self-esteem, and self-efficacy [42,43,44]. Yet, few studies have examined whether prosociality interventions such as SEL programs boost subjective well-being (i.e., happiness) or enhance other key positive psychological outcomes in primary school-aged children [44,45].

Second, there is mixed evidence that prosociality-focused or closely related programs can meaningfully impact children’s well-being. Some examinations of “positive education programs” in schools suggest that interventions promoting acts of kindness and other positive psychology activities (e.g., gratitude, developing core character strengths) can benefit children’s well-being (e.g., [46,47,48]). Yet, other studies indicate that prosociality-focused interventions are no more beneficial than an active control group. Thus, there is currently a limited and mixed understanding of how prosociality-focused interventions are associated with or impact primary school-aged children’s subjective well-being and positive psychological functioning overall. The present work aims to investigate whether participation in real-world prosociality programs in schools and community settings is associated with school-aged children’s well-being by examining two distinct, but theoretically overlapping programs: the Intergenerational Classroom and Cradles to Crayons.

### 1.2. Prosociality Programs: The Intergenerational Classroom and Cradles to Crayons

In the Intergenerational Classroom (iGen), approximately 25 students 11–12 years of age from the Saskatoon Public Schools District complete their 6th grade curriculum alongside senior care residents (referred to as “Elders”) in a local care home. Over the course of the school year, children are integrated into the Sherbrooke Community Centre, engaging in close and continued contact each day with senior residents, many of whom live with dementia and a variety of physical and intellectual conditions. These interactions facilitate daily opportunities for children to engage in both structured and spontaneous forms of prosocial behavior—from flipping the pages of a book for an Elder reading to the class to helping an Elder move from class to the cafeteria to have lunch. Throughout the school year, these daily prosocial activities may add up to help children develop an overall sense of purpose and meaning from their experiences with Elders—a potentially critical feature to unlock the positive effects of Intergenerational programs, including greater psychological well-being (see [34]).

Cradles to Crayons (C2C) is a non-profit organization that provides primary school-aged children with the opportunity to take a class field trip to package essential items for children living in poverty between birth and age 12. C2C collects children’s items (e.g., clothes, books, school supplies, toys) via community drives and corporate donations. These items are packed by volunteers in warehouses known as “Giving Factories” and distributed to local children who experience homelessness and low-income situations. Child volunteers spend part of their school day selecting and packaging the items that will be donated to another child in their community, giving children the chance to directly help someone in need over the course of a single afternoon.

We focus on these two interventions for both practical and theoretical reasons. Practically, we have a working relationship with these two organizations, which have been conducting their programs for the past 7 (iGen) to 19 years (C2C). While each organization has much anecdotal evidence for the efficacy of their program, neither program has been scientifically studied, leaving ample room for systematic examination. Theoretically, these interventions use distinct yet overlapping approaches to foster prosociality in children. Critically, rather than limit children’s experiences of prosociality to simple discussion and educational curriculum like some programs (e.g., [32]), both programs provide children with direct, hands-on opportunities to aid others, with interventions lasting from as little as several hours to as long as a 10-month school year.

Finally, both programs—either by design or facilitation—provide children with hands-on opportunities to help someone who may be quite different from themselves (e.g., older, differentially able, or from a different socio-economic status). It is often assumed that it may be easier or even more emotionally rewarding to help people who are like oneself. For example, some past work shows that school-aged children (aged 5–13 years) expect to be happier after helping similar others (e.g., [49]). On the other hand, there is good reason to suspect that helping people who are dissimilar to you might be particularly emotionally rewarding. Past work suggests that bridging social capital or establishing connections with people from heterogeneous groups (i.e., people who are socio-demographically different than oneself) can be valuable and rewarding for well-being and positive psychological functioning [50,51,52,53]. Thus, the present work offers an opportunity to examine whether any well-being benefits from engaging in prosocial behavior are different when helping similar (vs dissimilar) others.

### 1.3. Present Research

To examine the association between prosocial behavior and primary school-aged children’s well-being in the wild, we conducted two parallel studies—an exploratory correlational examination of the iGen program (Study 1) and a pre-registered experiment of C2C (Study 2). In Study 1, we surveyed students at the end of their year in iGen to explore whether children’s engagement in iGen and the meaning they derived from their primarily prosocial experiences with Elders were positively related to children’s well-being and positive psychological functioning. In Study 2, children reported their happiness before and after packaging supplies at a C2C Giving Factory for a child who experiences homelessness or financial hardship in the community. Critically, we randomly assigned volunteers to either prepare a package for a recipient who was similar or dissimilar in age and/or gender, allowing us to examine whether children were happier after helping than beforehand and whether happiness differed depending on who the child assisted.

## 2. Study 1: The Intergenerational Classroom (iGen)

### 2.1. Materials and Methods

#### Program Description

To participate in the Intergenerational Classroom (iGen) program, students across the Saskatoon public school district apply in their 5th grade year and a lottery is held to randomly select 25 participants from this large pool of applicants (approximately 150). In iGen, like all students in the 6th grade, students are taught the standard curriculum. However, each day during the school year, students interact closely with the senior care residents—colloquially referred to as “Elders”—who help facilitate the students’ learning alongside the teachers and Sherbrooke Community staff. Critically, across these interactions, students engage in both structured and spontaneous forms of prosocial action. For example, students can help Elders plant vegetables and flowers in the community garden or participate in “Coffee Club,” in which students give a presentation about different coffees from around the world and then make and serve coffee to the Elders. Of course, these activities can help students develop practical skills such as gardening or to learn about different cultures through the medium of coffee. However, these programs also enable students to engage in prosocial action by creating a much-appreciated community garden with and for the Elders or by spending time to make and serve them coffee.

Beyond the more structured prosocial activities, students engage in more spontaneous forms of prosocial action as they spend time having conversations with Elders or undertaking activities with them such as reading, chess, music/dance, and painting. To facilitate these interactions and activities, students learn to respectfully interact with and aid the Elders—many of whom live with dementia or varying physical and intellectual conditions. As such, students often help Elders experiencing physical disability to move around the center and help Elders experiencing cognitive impairments to engage in cognitively complex activities such as puzzles. Moreover, students learn to offer empathy and emotional support as they engage in conversations with Elders, offering friendship and support to those Elders who feel lonely, bored, or helpless. Thus, iGen provides a long-term immersive experience enabling students to engage in frequent prosocial acts with and for their Elders.

Finally, in addition to more hands-on experiences with prosocial action, the students often observe the teachers, staff, and Elders similarly undertaking prosocial interactions. For example, the teacher, staff, and Elders often demonstrate patient and active listening when speaking with Elders or offer help to Elders with ambulatory or cognitive challenges. This prosocial modeling ensures that Elders feel both respect and care while the students learn how to engage with Elders in a prosocial manner. As such, students also observe and learn from adult models as they socialize and normalize prosocial action across the daily interactions in the care home.

### 2.2. Participants and Procedure

In June 2020, we invited the 25 students who participated in iGen during the 2019–2020 academic year to complete a survey about their experiences in the program in exchange for an entry into a raffle to win one of four $25 Bookstore gift cards. Each of these students had applied to participate in the iGen program during the 2018/2019 academic year and were selected at random from a large pool of applicants (roughly 150) from various schools across Saskatoon to participate in iGen. Notably, three months before the end of the program, the COVID-19 pandemic declared in March 2020 disrupted the iGen activities as physical distancing rules prevented students from visiting Sherbrooke or interacting with one another in person. As such, our surveys were distributed online and primarily assess students’ experiences at Sherbrooke between September 2019 and March 2020 (see General Discussion for details).

We obtained parental consent and student assent in accordance with the data collection protocols approved by our institutional review boards. In total, 24 of the students (96%) completed the survey (50% female; *M*_Age_ = 11.5 years old; *SD*_Age_ = 0.51; race/ethnicity data were not collected); we received a partial response from one student (missing data on approximately 60% of the measures), but all available data were retained in analyses. In the survey, we captured students’ (1) experiences engaging with the Elders and how meaningful these experiences were perceived to be, (2) subjective well-being and positive psychological functioning, and (3) intentions to engage in prosocial acts in the future. These measures are described in detail below (see Table 1 for a summary of study measures, including sample items, scale anchors, and internal consistency statistics). Unless otherwise noted, we averaged across scale items to create an overall index of each outcome. The survey can be found on the Open Science Framework (OSF; https://osf.io/qb9gu/?view_only=b4215702b293408498a77c785340c5a9; accessed on 16 June 2021). 

#### 2.2.1. iGen Experiences and Meaning 

We captured students’ experiences and perspectives about interacting with the Elders of iGen using several measures. Specifically, students indicated which activities they regularly completed with the Elders from a list of five options (e.g., conversation, games/puzzles, art) and how often they undertook each of these activities during a regular week. The list of activities was generated in partnership with the director of iGen to be the most common activities that students could engage in during iGen. Students also had the option to list and report on the time they spent engaging in other activities, and also reported how much time they spent having conversations with Elders on a regular school day using a single face-valid item.

Importantly, we measured students’ overall sense of meaning from their interactions with Elders. While this measure does not directly capture the number of prosocial actions enacted over the school year, we reasoned that this measure offered initial, indirect insight into whether real-world prosocial behavior is associated with students’ psychological well-being because (1) prosociality is deeply integrated into the program and represents a core aspect of many of the student–Elder interactions, (2) people often find meaning from prosocial acts [5,7,66], and (3) a sense of meaning may be a critical component of any well-being benefits of Intergenerational programs [34]. Thus, the impact of regularly engaging in prosocial interactions with the Elders through iGen may be approximately captured through the meaning students derived from their interactions. We captured students’ perspectives about their time spent with the Elders and the meaning they ascribed to their conversations, activities, and experiences with the Elders across 12 items [54]. Specifically, students rated how valuable and precious these experiences were and to what extent they played an important role in some broader picture.

Next, students completed several face-valid items assessing how many lasting friendships they made with the Elders during the year, how interested they were in their experiences with the Elders, and their general impressions of the overall program (see Table 1). Finally, students completed three open-ended questions where they could describe whether and how participating in iGen changed the way they lived their lives or think about others—both generally and during the COVID-19 pandemic. These qualitative data are beyond the scope of the present analyses and will thus not be discussed in further detail here.

#### 2.2.2. Well-Being and Positive Psychological Functioning 

Students rated their well-being and positive psychological functioning on several validated measures. To capture well-being, students reported their life satisfaction, using both a single-item measure [55] and across nine items adapted from the Student’s Life Satisfaction Scale [56]; we report the analyses on both the single item and on the average of the SLSS items separately. Additionally, students rated their current vitality using six items adapted from the Subjective Vitality Scale [57]. Students also completed several measures of positive psychological functioning. Specifically, students reported their feelings of self-worth on the 10-item Rosenberg Self-Esteem Scale [58] and their sense of social-connectedness via 10 items that assessed social satisfaction and had high factor loadings as reported by Asher and colleagues [59]. Additionally, students rated their sense of self-efficacy and optimism using nine items from the Resilience Scale [60] and four items from the Life Orientation Scale-Revised [61], respectively.

#### 2.2.3. Intentions to Engage in Prosocial Action 

To assess whether students’ experiences and sense of meaning from iGen were associated with greater prosocial intentions, we measured students’ intentions to perform four prosocial acts in the future. Specifically, students rated the likelihood that they would: (1) keep engaged with Elders in their community, (2) volunteer their time, (3) donate to charity, or (4) give gifts to others in the future; these face-valid items were averaged to create an overall index of future prosocial intentions. We also evaluated students’ prosocial identity—the extent to which they viewed themselves as people who care about and act in ways that benefit others—with 10 items adapted from the Moral Identity Scale (e.g., “I am helpful” [62]). Moreover, we assessed students’ sense of prosocial motivation—their desire to help others in a meaningful way—using three items adapted from the Perceived Prosocial Impact scale [63]. Finally, we measured cognitive and affective levels of students’ empathy, which we averaged across the 13 items to create an overall index of empathy. Specifically, students rated their general tendency to take the perspectives of others and show concern for others’ emotions and well-being [64] in addition to their motivations to improve others’ well-being over the past school year [65].

### 2.3. Statistical Analysis

We used *SPSS* [67] to perform our correlational analyses and one-sample *t*-tests described below; we used two-tailed tests and the standard threshold for significance (*α* = 0.05). We tested the assumption of bivariate normality by visually inspecting the Q-Q plots of the marginal distributions of our key variables and by utilizing the Shapiro–Wilk test of normality. Given that violations of normality do not inflate Type I error but can reduce statistical power when using Pearson correlations, when we encountered evidence of non-normality, we followed recommended practice and estimated correlations using bootstrapping with 1000 resamples and bias-corrected accelerated confidence intervals [68,69]. Visual inspection of scatterplots revealed no evidence of non-linearity nor heteroscedasticity. Finally, we visually probed for potential outliers in our data using boxplots and report our results both with and without outliers.

### 2.4. Results

All data and code for Study 1 are available on the OSF at: https://osf.io/qb9gu/?view_only=55f711ce3bfc48a5bc863a416e9b42f8 (accessed on 16 June 2021).

#### 2.4.1. iGen Activities

First, we examined the extent to which students engaged in close and continued intergenerational contact throughout the school year (see Table 2 for descriptive statistics on these and all other measures). On average, students reported that they engaged in between three to four activities with the Elders during a regular week of the iGen program (*M* = 3.54; *SD* = 1.67). Moreover, students reported engaging in conversations with Elders approximately five times during a regular week (*M* = 4.91; *SD* = 1.14) and spending some-to-most of their time having conversations with Elders on any regular school day (*M* = 2.33; *SD* = 0.55). Beyond simple conversation, students tended to read stories with Elders between two and three times a week (*M* = 2.52; *SD* = 2.30) and engage in various activities (e.g., playing games/puzzles, engaging in art, engaging in music and dance) between one and two times a week (see Table 2). Thus, the iGen program regularly facilitated close and active contact between students and Elders each week of the program. 

We conducted exploratory analyses to examine whether the average amount of time students engaged in iGen activities during a typical week would differ between boys and girls. We found no gender differences in how often students engaged in art, music/dance, or had conversations with Elders (*p*s > 0.222). Interestingly, compared to girls, boys reported spending more time on average in a typical week playing puzzles/games (*M*_boys_ = 2.58, *SD*_boys_ = 1.38 vs. *M*_girls_ = 0.92, *SD*_girls_ = 0.669), *t*(15.9) = 3.78, *p* = 0.002, *d* = 1.60, 95% CI [1.18, 2.01], as well as reading stories (*M*_boys_ = 3.33, *SD*_boys_ = 2.31 vs. *M*_girls_ = 1.08, *SD*_girls_ = 1.88), *t*(22) = 2.62, *p* = 0.016, *d* = 1.12, 95% CI [0.31, 1.92]). 

#### 2.4.2. iGen Perceptions and Experiences of Meaning

Given that iGen encouraged close and continued intergenerational contact, we next examined students’ perceptions about their experiences, conversations, and activities with Elders. Overall, students reported having a very positive experience participating in iGen. On average, students reported being very interested in their conversations and activities with Elders (*M* = 6.21; *SD* = 0.58) and that they made between four and five long-lasting friendships with Elders (*M* = 4.58; *SD* = 3.20). Critically, students reported that they felt their conversations, experiences, and activities with Elders to be highly meaningful (*M* = 5.71, *SD* = 1.18). Indeed, a one-sample *t*-test revealed that students reported levels of meaning that were significantly above the midpoint of the scale, *t*(23) = 7.12, *p* < 0.001. Despite the overall high levels of meaning expressed by students, bivariate correlation analyses suggested that the more time that students spent engaged in activities with Elders, the more meaningful they reported their experiences to be, *r*(21) = 0.56, *p* = 0.006 (see Table 3). Taken together, these results suggest that students perceived their experiences in iGen to be highly positive and meaningful, and that the more engaged they were with the program, the more rewarding it was perceived to be.

#### 2.4.3. Meaning Derived from iGen and Well-Being

Given that the meaning students derived from their experiences with the Elders may roughly capture students’ perspectives on their primarily prosocial intergenerational interactions, we next explored how students’ experiences and perceptions of the iGen program were related to their reported well-being and positive psychological functioning. Overall, students reported very high levels of well-being and positive psychological functioning, with one-sample *t*-tests showing averages all significantly above the midpoint of the scales (see Table 2).

Critically, as shown in Table 2, the meaning that students derived from their experiences in the iGen program was strongly associated with most measures of well-being and positive psychological functioning (*r*s: 0.52–0.73); these correlations remained significant when accounting for potential outliers, though the magnitude of the correlations attenuated slightly (*rs*: 0.45–0.63). Students who derived more meaning from the program also reported higher levels of subjective vitality, self-worth, social connection, optimism, and self-efficacy. These correlations remained robust even when controlling for the number of long-lasting friendships that students made with Elders in the program. This casts doubt on an alternative possibility that simply making friends with Elders is driving the relationship between meaning and well-being. Rather, what students may find more meaning from is that they could be of use to and help the Elders in the Sherbrooke community. Interestingly, while we observed a significant correlation between meaning and our single-item measure assessing overall life satisfaction (*r* = 0.54, *p* = 0.007), meaning was not significantly related to life satisfaction as captured by the SLSS (*r* = 0.34, *p* = 0.118). Moreover, when we removed potential outliers, the correlation between meaning and our single-item life satisfaction measure was reduced, *r*(20) = 0.23, *p* = 0.303, possibly due to the reduced sample size. Taken together, these results suggest that the meaningful, close, continued, and primarily prosocial intergenerational contact that students had through iGen may positively shape students’ subjective well-being and positive psychological functioning.

#### 2.4.4. Meaning Derived from iGen and Prosociality

Finally, we explored whether students’ experience of meaning in the program was associated with higher levels of reported prosociality. As with their well-being reports, students rated themselves very highly on all our measures of prosociality, with averages all significantly above each scale midpoint (see Table 2). Despite the overwhelmingly positive ratings for both the program and students’ prosociality, we again found large positive associations between the meaning students derived from the program and their prosociality levels (*r*s: 0.59–0.84); as above, these correlations remained significant when several potential outliers were removed (*r*s: 0.51–0.64). Indeed, students who derived more meaning from their experiences with Elders were more likely to want to engage in future prosocial acts such as spending time with Elders in their community or donating their time and money to charity (*r* = 0.73, *p* < 0.001). Similarly, students who experienced the greatest meaning from participating in iGen were also more likely to rate themselves as helpful and generous people, to hold greater motivations to help others, and to be more empathetic (see Table 2). Overall, these results are consistent with the hypothesis that regular experiences of meaningful—often prosocial—intergenerational contact may positively shape students’ prosociality development.

### 2.5. Discussion

Study 1 suggests that participation in real-world school programs, such as iGen, that model and facilitate meaningful, prosocial intergenerational contact between children and Elders is associated with greater well-being, psychological functioning, and prosocial intentions among children. Indeed, despite very high levels of well-being and prosociality among iGen students, we observed a positive association between meaningful participation and various measures, such as life satisfaction, vitality, social connection, empathy, and prosociality. Therefore, these findings are consistent with the possibility that program engagement is related to greater positive psychological functioning. It is encouraging to see that the relationships between meaning and well-being/prosociality remained when controlling for the friendships students made with Elders, suggesting that students may have found their prosocial actions to be the most meaningful aspect of their interactions in iGen.

The iGen program is an immersive 10-month long program spanning the academic year, which provides youth with several opportunities to foster meaningful connections and practice prosociality. Could participation in a shorter prosocial program also be associated with greater well-being? To examine this question in a parallel study, we assessed happiness before and after children volunteered for two hours in a C2C Giving Factory—as part of a classroom field trip or outing with their parents or sports team—where they packaged essential items for a local child who experiences poverty. Critically, child volunteers were randomly assigned to either prepare a package for a similar child (i.e., someone of their gender and similar age) or a less similar child (i.e., someone several years older or younger and/or a different gender).

The design of Study 2 complements and improves upon Study 1 by allowing us to investigate two key additional questions important for the study of prosociality. First, we could examine whether children reported any changes in happiness from before to after engaging in a relatively brief, real-world act of generosity. As such, we were able to control for baseline levels of happiness. Importantly, rather than approximate prosocial engagement from the meaning students derived from their prosocial interactions as in Study 1, we could establish a more direct link between real-world prosocial behavior and happiness to help eliminate alternative explanations. Second, our experimental design permitted us to probe whether their giving to a similar (vs. dissimilar) other was associated with greater happiness gains. In light of past research suggesting that people may enjoy helping similar others more than dissimilar others [49], we pre-registered that children would be happier after volunteering than before and that children who helped someone who was socio-demographically similar would be happier than children who helped someone who was dissimilar.

## 3. Study 2: Cradles to Crayons (C2C)

### 3.1. Materials and Methods

#### Program Description

Cradles to Crayons (C2C) provides essential items (e.g., shoes, books, clothing, supply packs) to disadvantaged children from birth to 12 years old. Items are often packaged by student volunteers during 2 h field trips with their class, sports team, or parents to the C2C warehouses, known as “Giving Factories.” Children as young as five years old can volunteer in Giving Factories, which are in Boston, Chicago, and Philadelphia (USA).

### 3.2. Participants and Procedure

Between June 2018 and August 2019, 257 children participated in the study during their field trips to visit C2C. Parents provided informed consent and all data collection procedures were approved by our Institutional Review Board. In line with our pre-registered exclusion criteria, we excluded seven children for not reporting their pre- or post-volunteering happiness; an additional 11 children were excluded because their permission forms could not be verified. After removing these 18 participants, we were left with a final sample of 238 children (55% female; *M*_Age_ = 8.65 years; *SD*_Age_ = 2.07, range = 5–12; race/ethnicity data were not collected). Our planned sample size was *N* = 630 with an interim analysis at *N* = 400 to detect a small effect (Cohen’s *d* = 0.20) with 80% power using a one-tailed, independent samples *t-*test at *α* = 0.05. However, data collection in 2020 was disrupted by COVID-19. We registered an update on the OSF that our final sample was *N* = 239. However, we failed to notice until after posting the document that one of the participants had only been four years of age and thus was ineligible to volunteer in the Giving Factory; excluding this participant left us with a sample of *N* = 238. Procedural details and analytical plans for this study were pre-registered on the OSF at: https://osf.io/6d38e/?view_only=3dd2520007254f68b6eb22f31cedf7a6 (accessed on 29 July 2019).

Upon arriving at the C2C Giving Factory, participants gave informed consent and reported their child’s gender. Afterward, participants received an orientation from C2C staff members to learn how to inspect, sort, and package donations and to learn more about the study. During the orientation, children were familiarized with a six-point scale used to rate feelings of happiness (see Figure 1). The scale ranged from “really not happy”—represented by a large frowning face and the number one—to “really happy”—represented by a large smiling face and the number six; see OSF for sample script used to explain the scale. Following the orientation, children were given a printed version of the happiness scale and were asked to indicate their pre-volunteering happiness (i.e., “How do you feel right now?”).

Participants were then randomly assigned to either one of two conditions: the *Similar Recipient* condition (*n* = 95; 51% female; *M*_Age_ = 8.32 years; *SD*_Age_ = 2.11, range = 5–12) or the *Dissimilar Recipient* condition (*n* = 143; 58% female; *M*_Age_ = 8.87 years; *SD*_Age_ = 2.01, range = 5–12). In the *Similar Recipient* condition, participants were asked to prepare a package for a child of the same gender and within two years of the participant’s age. Meanwhile, in the *Dissimilar Recipient* condition, participants were asked to prepare a package for a child who was a different gender and/or more than two years older or younger than the participant. Participants received a packing slip that introduced the recipient (i.e., recipient’s name, age, gender, zip code, and caregiver’s name) and then packaged the requested items (e.g., size 7 everyday shoes, books, hygiene kit, art supply pack). Afterward, children indicated their post-volunteering happiness on a second printed version of the same happiness scale (see Figure 1).

Finally, participants were asked if they wanted to visit C2C again (yes or no), if they remembered the gender and age of their recipient, and if they knew someone like the child on their form (yes or no). The caregivers of pre-literate children read each of the last three questions to the participant and recorded the child’s initial response without additional prompts or questions.

### 3.3. Hypotheses and Deviation from Pre-Registration

We pre-registered two main predictions. First, we pre-registered that children’s reported happiness would increase from pre-intervention to post-intervention. Critically, we predicted that this increase in happiness would be greatest for volunteers assigned to the *Similar Recipient* condition (i.e., children who packaged items for recipients similar in age and gender) than volunteers assigned to the *Dissimilar Recipient* condition (i.e., children who packaged items for recipients of a different gender and larger age difference). To test whether changes in happiness differed by condition, we pre-registered two complementary analyses using one-tailed tests. Specifically, we indicated that we would regress participants’ post-intervention happiness reports on condition (0 = *Dissimilar Recipient* condition; 1 = *Similar Recipient* condition) while controlling for participants’ pre-intervention happiness to see whether any changes in happiness depended upon condition. Further, we pre-registered that we would conduct an independent samples *t*-test to examine whether post-intervention happiness would be greater for participants in the *Similar Recipient* condition as compared to participants in the *Dissimilar Recipient* condition. However, since finalizing our pre-registration, we learned that a repeated measures ANOVA would more robustly test whether the average rate of change in happiness from pre- to post-intervention differed by condition. Thus, where applicable, we report the findings from these additional tests alongside our pre-registered analysis plan.

Second, we pre-registered our prediction that children would be more willing to volunteer at C2C again when selecting items for a similar (vs. dissimilar) recipient and that this would depend upon condition assignment, post-intervention happiness, and/or indices of demographic similarity between the participant and recipient (i.e., age, gender). However, upon examination of the data, 99% of participants (*N* = 232) indicated that they were willing to return to C2C in the future. As such, we do not report these analyses here given the lack of variability in responses.

### 3.4. Statistical Analysis

We used *SPSS* [67] to perform our analyses; we used one-tailed tests for directional hypotheses, two-tailed tests for non-directional hypotheses, and the standard threshold for significance across tests (*α* = 0.05). We tested for normality of our variables and model residuals by visually inspecting Q-Q plots and utilizing the Kolmogorov–Smirnov tests of normality. Given that paired-samples *t*-tests, OLS regression analyses, and repeated measures ANOVAs are generally robust to violations of normality [70], we did not perform any transformations on our data when we found evidence of non-normality. Visual inspection of scatterplots revealed no evidence of non-linearity nor heteroscedasticity. Finally, we visually probed for potential outliers in our data using boxplots. In our OLS regression, we additionally probed for outliers in our outcome by comparing externally studentized residuals against a critical value associated with a *t*(236) distribution and looked for potential globally influential outliers by examining DFFITS and DFBETAS against a critical value of one [71]. Removing potential outliers did not change the interpretation of results; thus, we report the analyses including the full sample to maximize statistical power.

### 3.5. Results

All data and code for Study 2 are available on the OSF at: https://osf.io/m97vx/?view_only=1ad6a4650f524b999c5fcb042cf26e5d (accessed on 28 April 2021).

#### Pre-Registered Analyses

As pre-registered, we first tested whether participants’ self-reported happiness increased from pre-intervention to post-intervention for the overall sample and within each condition separately. As predicted, a series of paired samples *t*-tests revealed that children’s self-reported happiness increased from pre-volunteering (*M* = 5.18, *SD* = 0.83) to post-volunteering (*M* = 5.63, *SD* = 0.56), *t*(237) = 9.32, *p* < 0.001, *d_p_* = 0.63, 95% CI [0.49, 0.77]. The happiness boost was observed in both the *Dissimilar Recipient* condition, *t*(142) = 7.54, *p* < 0.001, *d_p_* = 0.69, 95% CI [0.49, 0.89], and the *Similar Recipient* condition, *t*(94) = 5.47, *p* < 0.001, *d_p_* = 0.54, 95% CI [0.33, 0.75]; see Figure 2. Thus, it appears that spending time packaging essential items for a child who experiences homelessness and/or poverty is associated with greater happiness in primary school-aged children, regardless of whether the target is similar or dissimilar to one’s identity. While this result is consistent with past work, we did not include a non-prosocial activity control group in our design; thus, we cannot rule out the possibility that children simply became happier over time (see General Discussion).

We next tested our prediction that the change in happiness from pre-intervention to post-intervention would be greater for participants in the *Similar Recipient* condition as compared to the *Dissimilar Recipient* condition. In contrast to our directional prediction, regression analyses revealed that condition was not associated with post-intervention happiness while holding pre-intervention happiness constant, *b* = −0.09, *SE* = 0.06, 90% CI = [−0.20, 0.02], *β* = −0.08, 90% CI [−0.19, 0.02], *t*(235) = −1.43, *p* = 0.923. Similarly, independent *t*-test analyses revealed no significant mean differences in post-volunteering happiness between participants in the *Similar Recipient* (*M* = 5.57, *SD* = 0.60) and *Dissimilar Recipient* conditions (*M* = 5.66, *SD* = 0.53), *t*(185.98) = 1.27, *p* = 0.897, *d* = 0.17, 90% CI [−0.05, 0.39]. Finally, converging with the former two tests, a repeated measures ANOVA revealed that the mean change in happiness from pre- to post-intervention did not differ by condition, *F*(1, 236) = 0.75, *p* = 0.389, *η*_p_^2^ = 0.00, 90%CI [0.00, 0.03]. Therefore, these tests suggest that being assigned to package essential items for a recipient of similar age and gender was not associated with greater well-being for children than packaging items for a dissimilar recipient. (Note: We additionally examined whether age moderated the effect of condition on the average change in happiness from pre- to post-intervention. An exploratory repeated measures ANOVA revealed that age was neither individually associated with nor moderated the effect of condition on the pre- to post-intervention change in well-being (*p*s > 0.073).)

Broadly, the previous analyses focused on examining how being assigned to give to similar (vs. dissimilar) recipients impacted children’s post-intervention happiness. However, whereas participants in the *Similar Recipient* condition were always the same gender and aged within two years of each other, participants in the *Dissimilar Recipient* condition could differ from the recipient in terms of gender, age, or on both dimensions. Thus, we also pre-registered our intention to examine whether changes in happiness from pre- to post-intervention differed as a function of more nuanced conceptualizations of dissimilarity. Specifically, we explored whether the change in pre- to post-volunteering happiness was related to either a match or mismatch in self-reported gender and/or three different operationalizations of age difference: (1) the absolute difference in age between participant and recipient, (2) whether the recipient was older or younger than the participant, and (3) whether the participant and recipient were the same age, similar in age (±one year), or disparate in age (±two or more years). We found no consistent evidence across various tests and various analytic strategies testing whether the more nuanced conceptualizations of similarity (vs. dissimilarity) were associated with greater post-volunteering happiness (see Appendix A for details). Taken together, these findings coincide with the analyses above and (contrary to our predictions) suggest that volunteering to help a dissimilar recipient correlated with well-being similarly as volunteering to help a recipient like themselves.

### 3.6. Discussion

Converging with Study 1 and past lab experiments (e.g., [8,11]), the findings of Study 2 suggest that school-aged children who took a field trip to participate in a prosocial program in the community reported increases in subjective well-being from before to after the task. In contrast to our pre-registered predictions, we found no evidence that volunteering at C2C was related to well-being any differently when children packaged essential items for a child who experiences homelessness and/or poverty of similar rather than dissimilar gender and/or age.

## 4. General Discussion

Across two studies, we examined whether engagement in two real-world prosociality programs was associated with greater psychological well-being in primary school-aged children. Using both a correlational survey and a pre-registered experiment, we saw converging evidence supporting this hypothesis. Notably, involvement in each program was associated with greater well-being despite differences in how much time children spent engaging in prosocial action—from a single afternoon to across an entire school year.

The present findings align with previous research demonstrating an association between prosocial behavior and greater psychological well-being [8,9,11]. As such, it is possible that previous research documenting the relationship between prosocial behavior and well-being may generalize to real-world contexts, despite the literature’s prioritization of internal over external validity. Given the scientific community’s recent push to conduct more generalizable research and to collect larger samples (e.g., [72,73,74,75]), the present work underscores the importance of testing questions in real-world community settings. Of course, the unique nature and challenges often associated with examining real-world programs (e.g., [74])—particularly amidst a global pandemic—limited our capacity to establish strong causal conclusions (see below). However, we believe our work contributes to efforts for more rigorous, generalizable research examining the emotional benefits of school- and community-based prosocial behavior in the wild and offers valuable guidance for researchers to conduct additional fieldwork.

Study 1 examined an immersive program that created a long-term, in-depth experience for primary school-aged children to develop their prosociality. Of course, the immersive nature of the program limits its capacity to only 25 students a year, which makes conducting a well-powered experiment on this program with a representative group of students particularly challenging. As a result, Study 1 relied on a correlational design and a small, unrepresentative sample, meaning we cannot determine whether participating in iGen *causes* changes in psychological well-being, nor can we eliminate possible self-selection biases or determine if these results generalize to children of various socio-demographic contexts.

Happier and more helpful students may be more likely to enroll in prosocial programs, such as iGen, and our data suggest that this may be the case. Students in Study 1 reported high levels of prosociality and well-being that were significantly above the midpoint of the scales. However, it is important to note that the iGen program attracts significant attention from the Saskatoon community, with ~150 applicants annually for only 25 spots. Thus, given the random selection process, our sample likely reflects the interests and well-being of the larger and broader applicant pool. Moreover, it is encouraging to see that levels of engagement with and meaning derived from participating in iGen the program were associated with psychological well-being, even among this sample reporting high levels of well-being. This finding is notable given that the restricted range and variability on our measures are likely to reduce the possibility of detecting statistically significant associations.

Critically, future work could capitalize on the natural experimental context provided by iGen’s lottery selection process and should include a more direct measure of prosocial actions completed in the program. Specifically, researchers could broadly survey students across the school district—including those randomly selected to participate in iGen, those who applied but were not selected, and those who did not apply to iGen at all—both before and after the iGen program. This design would help to move closer to a causal test of program participation, better account for self-selection biases, and provide a baseline measure of well-being to assess overall levels of change in well-being. Further, including a more direct measure of prosocial engagement in the program rather than an approximation through the meaning derived from the program will help eliminate alternative explanations. More broadly, future studies should aim to recruit larger samples so that additional interesting questions can be explored, such as whether there are gender differences in the types of helping enacted or the emotional rewards of kindness.

Notably, we examined the iGen program during a particularly strange time—roughly seven months into iGen, the COVID-19 pandemic was declared. To keep the community safe, students stopped attending Sherbrooke and completed their final three months of iGen individually, from home. Though technology connected students and Elders virtually through Zoom a handful of times, the iGen experience was considerably muted for the final third of the program. As such, our study provides a conservative examination of iGen and the strong correlations we observed between students’ truncated experiences and their well-being are encouraging as they suggest that there may be great potential for the program at full capacity. More work assessing iGen under standard conditions is needed. However, for now, the lives and capacities of the students, teachers, and administrators at schools and Elders and staff at long-term care homes around the world have been disrupted due to the pandemic and will continue to be for the foreseeable future (e.g., [76,77]). We hope our work will better equip future researchers to conduct much-needed additional fieldwork not feasible during the pandemic. Beyond additional quantitative work, our data also suggest iGen may be a fruitful context by which to conduct qualitative methods. Through focus groups and qualitative interviews, researchers can more deeply examine the meaningful relationships of students and Elders to gain richer insights into the link between prosocial acts and well-being in the wild [78].

In Study 2, we offset some of the limitations of Study 1 by conducting a pre-registered experiment with a much larger sample. Despite these methodological advantages, the experimental design did not include a control condition to help eliminate important alternative explanations. Indeed, in Study 2, there was an increase in happiness across both conditions from before to after preparing gifts for children who experience homelessness and/or poverty. It is difficult to rule out the possibility that this increase in happiness is simply due to the passage of time or social contact; it could be that children are becoming happier over the course of their afternoon together. For example, other studies have noted an increase in children’s well-being from pre- to post-assessment both among students who engaged in kind acts and an active control condition and thus fail to demonstrate that any gains in well-being are due to prosocial action (e.g., [19]). Thus, future researchers would do well to include a control condition in which children perform a non-generous task as participants in the other condition packaged gifts for children who experience homelessness and/or poverty in the community. That being said, the findings of Study 2 align with previous work which suggests that people tend to feel happier after prosocial acts (e.g., [8,9,11]), so it is plausible that giving to a range of recipients could boost happiness.

Interestingly, despite work suggesting that helping similar others may be perceived as more emotionally rewarding (e.g., [49]), our data suggest that there may be no discernable advantage to helping similar over dissimilar others. Indeed, as evidenced by the null effect in Study 2, those who helped others of the same age and gender reported similar levels of post-helping happiness and similar increases in happiness as those who helped someone unlike them in age and gender. These results challenge people’s intuitions that giving to similar others may be more rewarding. This is important because people in need may appear to be quite different from potential helpers (e.g., [79,80,81]). Critically, this work aligns broadly with the concept of bridging social capital which suggests that it is possible that giving to people who are socio-demographically dissimilar may be emotionally rewarding (e.g., [50,51,52,53]).

Just as field work examining prosociality-focused interventions can advance underlying theories of prosociality development and psychological well-being, the present work also underscores the practical importance of performing real-world investigations. Indeed, scientific practice would do well to continue to go beyond the lab and evaluate the value of real-world programs already in the community. For example, data show that parents, teachers, and communities broadly are interested in fostering prosociality and want their children to be happy [82]. Thus, examining prosociality-focused programs for primary school-aged children, such as iGen and C2C, can offer valuable guidance to communities and—to the extent that the present findings align with past work—help communities leverage these programs to both nurture prosociality and enhance the psychological well-being of the next generation.

Should our findings be replicable with additional rigorous experimental methods, the present field work can also provide initial information to practitioners and educators seeking to provide quality education to students around the world (e.g., [1,2,79,80,81]). For example, the immersive, long-term nature of iGen requires a lot of careful management, despite using essentially the same teaching and caring resources that would be needed if there were no mixing of elder care and education. Moreover, educational ministries and school and long-term care home administrators must consider the safety and feasibility of hosting Intergenerational programs. While programs such as iGen may be impactful and popular, there may be practical challenges to making such programs available more widely in school systems. Thus, evidence of the sort we reported here can strengthen the case for program expansion. On the other hand, given the relatively low time and financial costs associated with the C2C program, it may be much more amenable to the concept of scalability; it can accommodate dozens of children in an afternoon. Although C2C may not be as immersive as iGen, it may produce momentary benefits that motivate further prosociality. Indeed, to the extent that the positive feelings associated with giving to others reinforce the practice of giving, past work suggests that children may be motivated to recreate those feelings and either return to the Giving Factory or initiate other forms of prosociality in a positive feedback loop [9,83].

## 5. Conclusions

Encouraging the next generation to be more prosocial and happier are often desirable goals for people and nations around the world (e.g., [84,85,86,87,88]). However, people often have little direction on how to achieve these large, amorphous goals. Past work conducted in the lab offers internally valid evidence that helping others can boost happiness for very young children and adults, e.g., [9]. Meanwhile, a host of real-world, often costly school programs exist in which communities try to develop prosociality in children (e.g., [32,38,89,90,91]), but data are rarely collected in these real-world community settings or with school-aged children. Particularly following the COVID-19 pandemic, it is critical to bolster educational practices to teach meaningful skills that promote well-being over children’s lifetimes [1,2]. By going beyond lab-based experiments and examining the real-world association between prosocial behavior and well-being across two very different, but complementary school programs, the present studies show that prosocial behavior may be one meaningful tool to help build higher quality, sustainable, and happier education around the world.

## Figures and Tables

**Figure 1 ijerph-20-04403-f001:**
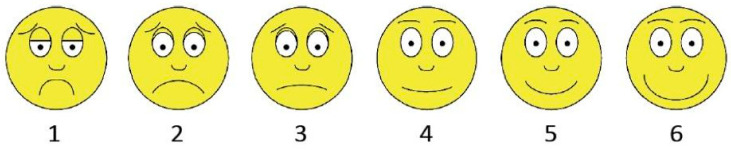
Scale used to capture momentary happiness before and after volunteering at Cradles to Crayons (C2C) Giving Factory (Study 2).

**Figure 2 ijerph-20-04403-f002:**
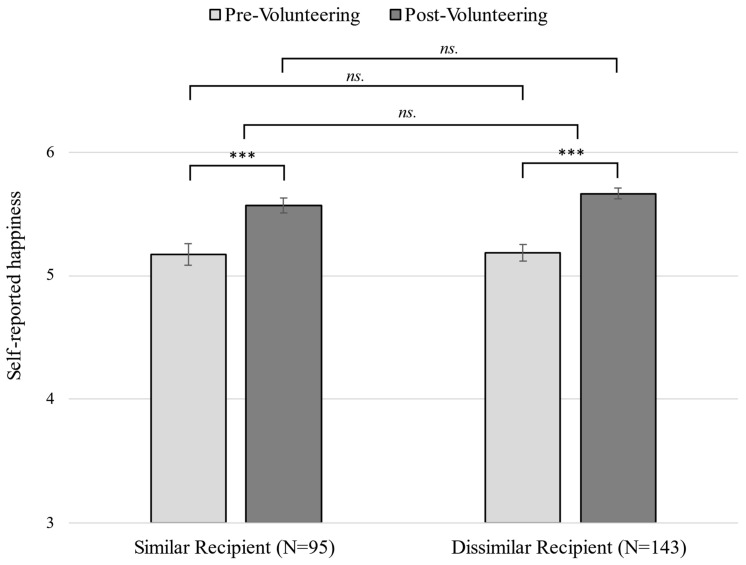
Self-reported happiness before and after packaging items for similar or dissimilar children at the Cradles to Crayons (C2C) Giving Factory. *** *p* < 0.001; *ns* = non-significant. Error bars represent standard errors.

**Table 1 ijerph-20-04403-t001:** Summary of measures (Study 1).

Measure	Source	N of Items	α	Sample Item(s)	Response Options
Experience of meaning	[54]	12	0.96	To what degree do you feel that your conversations, activities, and experiences with the Elders were… …meaningful …valuable	1—Not at all 7—Very much
Number of friendships	Face Valid	1	NA	Did you make any lasting friendships with the Elders of the Sherbrooke Community centre?	Number of friendships reported
Engagement in activities	Face Valid	1	NA	What kind of activities did you regularly do with the Elders? …Games/Puzzles …Art …Read Stories …Music/Dance …Conversation …Other (please specify)	Number of activities selected
Time spent engaging in activities	Face Valid	1	NA	During a regular week, how often would you do each of these activities with the Elders of the community centre? …Games/Puzzles …Art …Read Stories …Music/Dance …Conversation …Other (please specify)	0—Not at all 1—Once a week 2—Twice a week 3—Three times a week 4—Four Times a week 5—Five times a week 6—Six or more times a week
Life Satisfaction	Single Item [55]	1	NA	How satisfied are you with your life as a whole?	1—Dissatisfied 10—Satisfied
	Student’s Life Satisfaction Scale [56] ^a^	9	0.86	I like the way things are going for me My life is going well I have a good life	1—Never 2—Sometimes 3—Often 4—Almost Always
Subjective Vitality	Subjective Vitality Scale—Short [57] ^a^	6	0.91	Typically… …I feel energized …I feel alive and vital (full of energy, lively)	1—Not true at all 7—Very true
Self-worth	Rosenberg Self Esteem Scale [58]	10	0.75	On the whole, I am satisfied with myself I feel that I have a number of good qualities I take a positive attitude toward myself	1—Strongly Disagree 2—Disagree 3—Agree 4—Strongly Agree
Social Connection	[59]	10	0.89	It’s easy for me to make new friends at school. I have lots of friends	1—Not true at all 5—Always true
Self-efficacy	Resilience Scale [60]	9	0.91	When I am in a difficult situation, I can usually find my way out of it. I usually manage one way or another.	1—Not at all 7—Extremely
Optimism	Life Orientation Test-R [61]	4	0.80	In uncertain times, I usually expect the best. I’m always optimistic about my future	1—Strongly Disagree 5—Strongly Agree
Prosocial Intentions	Face Valid	4	0.86	How likely are you to… …volunteer your time in the future? … give gifts in the future? … donate to charity in the future? …keep engaged with Elders in your neighborhood or community in the future?	1—Not at all 7—Very much
Prosocial Identity	Moral Identity Scale [62] ^a^	10	0.91	I try to help others whenever I can. (I am helpful.) I am giving towards others (I am giving.)	
Prosocial Motivation	Prosocial Impact Scale [63] ^a^	3	0.72 ^b^	I try to have a positive influence on others I try to be a force of good.	1—Strongly Disagree 2—Disagree 3—Slightly Disagree 4—Neutral 5—Slightly Agree 6—Agree 7—Strongly Agree
Empathy	Empathy Scale [64,65]	13	0.87	[Perspective Taking] Before criticizing somebody, I try to imagine how I would feel if I were in their place [Empathic Concern] I often feel concerned for people who are less fortunate than me [Empathic Motivation/Compassion] It is important to me to help people who are less fortunate	1—Strongly Disagree 2—Disagree 3—Uncertain 4—Agree 5—Strongly Agree

Note. ^a^ Items were adapted to be more appropriate for children. ^b^ Our measure of prosocial motivation contained three items. However, the item “I try to make a difference” reduced scale reliability to unacceptable levels (*α* = 0.68). We removed the problematic item and conducted our analyses with the shortened, more reliable scale.

**Table 2 ijerph-20-04403-t002:** Item means and bivariate correlations between meaning, friendships, engagement, well-being, and prosociality.

	Mean (SD)	Meaning from Conversations, Activities, and Experiences with Elders ^a^ 1—Not At All 7—Very Much	Number of Friendships ^a^	Engagement in Activities (Sum) Max. 6	Engagement in Activities (Time) 0—Not at All 6—6+ Times/Week
Life Satisfaction (Single-item) ^a^ 0—Dissatisfied 10—Satisfied	**7.92** (1.89)	**0.54 ****	0.31	0.01	0.35
Life Satisfaction (SLSS) 1—Never 4—Almost always	**3.26** (0.47)	0.34	0.13	0.20	0.20
Vitality 1—Strongly Disagree 7—Strongly Agree	**5.18** (1.18)	**0.64 *****	0.21	0.21	**0.41 ***
Self-worth 1—Strongly Disagree 4—Strongly Agree	**2.91** (0.38)	**0.60 ****	−0.06	0.11	0.32
Social Connection 1—Not at all true 5—Always true	**3.89** (0.66)	**0.58 ****	0.18	0.10	**0.50 ****
Optimism 1—Strongly Disagree 4—Strongly Agree	**2.76** (0.60)	**0.52 ***	0.13	0.28	**0.41 ***
Self-efficacy ^a^ 1—Not at all 7—Extremely	**5.24** (1.15)	**0.73 *****	−0.20	0.05	0.41 ^†^
Prosocial Motivation ^a^ 1—Strongly Disagree 7—Strongly Agree	**6.04** (0.78)	**0.76 *****	0.01	0.41 ^†^	0.38 ^†^
Prosocial identity ^a^ 1—Not at all 7—Very much	**6.00** (0.81)	**0.84 *****	0.16	0.15	**0.47 ***
Empathy 1—Strongly Disagree 5—Strongly Agree	**4.09** (0.50)	**0.59 ****	0.15	0.14	0.34
Future prosocial behavior ^a^ 1—Not at all 7—Very much	**6.01** (0.87)	**0.73 *****	−0.02	0.07	**0.46 ***
Mean (SD)		**5.71** (1.18)	4.58 (3.20)	3.54 (1.67)	**1.98** (0.89)

Note. ^a^ Variables with a superscript (^a^) indicate the presence of significant non-normality (*W*s > 0.878, *p*s < 0.032); bootstrapping techniques with 1000 resamples were used to estimate correlation coefficients in the presence of non-normal data. All bias-corrected accelerated 95% confidence intervals aligned with the interpretations provided by the *p*-values and are thus not included in this table. ^†^
*p* < 0.10; * *p* < 0.05; ** *p* < 0.01; *** *p* < 0.001. Engagement in activities (sum) is equal to the total number of different activities students engaged in regularly with Elders. Bolded means are significantly different from scale midpoint.

**Table 3 ijerph-20-04403-t003:** Bivariate correlations between meaning, friendships, and engagement variables.

	Meaning from Experiences with Elders ^a^	Number of Friendships ^a^	Engagement in Activities (Sum)	Engagement in Activities (Time)
**Meaning from experiences with Elders ^a^**	1			
**Number of Friendships ^a^**	0.15	1		
**Engagement in activities (Sum)**	0.21	0.38 ^†^	1	
**Engagement in activities (Time)**	**0.56 ****	**0.45 ***	**0.61 ****	1

Note. ^a^ Variables with a superscript (^a^) indicate the presence of significant non-normality (*W*s > 0.856, *p*s < 0.003); bootstrapping techniques with 1000 resamples were used to estimate correlation coefficients in the presence of non-normal data. All bias-corrected accelerated 95% confidence intervals aligned with the interpretations provided by the *p*-values and are thus not included in this table. ^†^
*p* < 0.10; * *p* < 0.05; ** *p* < 0.01. Engagement in activities (sum) is equal to the total number of different activities students engaged in regularly with Elders. Bolded means are significantly different from scale midpoint.

## Data Availability

All materials, data, and code can be found on the Open Science Framework (OSF) at https://osf.io/qb9gu/?view_only=b3d4d5d614e84efaac7871d317ed60fb (accessed on 16 June 2021) for Study 1 and at https://osf.io/fchwe/?view_only=bf9e9639f53c4697ac910c98600d00ab (accessed on 15 May 2018) for Study 2.

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
