# Peer review of "Are Real-World Prosociality Programs Associated with Greater Psychological Well-Being in Primary School-Aged Children?"

_ijerph, 2023, doi:10.3390/ijerph20054403_

Round 1

Reviewer 1 Report

Thank you very much for the opportunity to read and review the manuscript entitled “Teaching Prosociality and Well-Being Beyond the Classroom: Are Real-World Prosociality Programs Associated with Greater Psychological Well-Being in Primary School-Aged Children?”.

The paper aims to investigate whether engaging in real-world prosociality programs in schools and community settings is associated with primary school-aged children’s well-being by examining two distinct programs, lasting an entire school year and an afternoon respectively.

Even if the subject of the paper is interesting and well discussed, some aspects can be clarified or improved to enhance the quality of the manuscript.

Detailed comments are reported below:

Abstract

The abstract is concise and easy to read.

Introduction

In general, the rationale of the study is convincing, well discussed and argued.

Bibliographical references are recent and appropriate.

-          At the beginning of paragraph 1.2, I suggest making explicit the age of children attending sixth grade.

Study 1

-          In paragraph 2.1.1 I suggest describing more clearly the research hypotheses and the relationships between the variables analysed. Specifically, the authors consider variables related to prosocial experience as predictors, but the analyses presented below are correlational. This point needs to be clarified.

-          The title of Table 2 reports “Item Means and Bivariate Correlations Between Predictors and Well-Being Outcomes”. The title is incomplete, as the Table contains also Prosociality Outcomes. Please modify the title.

-          In paragraph 2.4.4 Authors state: “Finally, we explored whether students’ experience of meaning in the program predicted higher levels of reported prosociality”. Since the analyses are correlational, I suggest not referring to predictive and causal hypotheses.

-          In paragraph 2.5 I suggest that more space should be devoted to the discussion of study 1 rather than the introduction to study 2.

Study 2

-          In paragraph 3.2 the SDAge value is incomplete. Please check.

Discussion and Conclusion

Discussion, Conclusions and Limitations are well argued.

As a limitation, I suggest to refer also to the fact that gender differences were not considered. it would be interesting to explore this further, as the literature reports significant gender differences in both prosociality and well-being.  

Author Response

  1. Thank you very much for the opportunity to read and review the manuscript entitled “Teaching Prosociality and Well-Being Beyond the Classroom: Are Real-World Prosociality Programs Associated with Greater Psychological Well-Being in Primary School-Aged Children?”. The paper aims to investigate whether engaging in real-world prosociality programs in schools and community settings is associated with primary school-aged children’s well-being by examining two distinct programs, lasting an entire school year and an afternoon respectively. Even if the subject of the paper is interesting and well discussed, some aspects can be clarified or improved to enhance the quality of the manuscript.

Response: We thank Reviewer 1 for this accurate and positive summary of our work. We also appreciate the suggestions that follow.

  1. The abstract is concise and easy to read.

Response: We are pleased to know that Reviewer 1 found the abstract concise and approachable.

  1. In general, the rationale of the study is convincing, well discussed and argued.

Response: We thank Reviewer 1 for this positive assessment of the introduction.

  1. Bibliographical references are recent and appropriate.

Response: We are pleased to know that Reviewer 1 was satisfied with the literature review and refences. 

  1. At the beginning of paragraph 1.2, I suggest making explicit the age of children attending sixth grade.

Response: As requested, we now explicitly identify the age of students in the sixth grade attending iGen in paragraph 1.2 (p. 4).

  1. Study 1.In paragraph 2.1.1 I suggest describing more clearly the research hypotheses and the relationships between the variables analysed. Specifically, the authors consider variables related to prosocial experience as predictors, but the analyses presented below are correlational. This point needs to be clarified.

Response: We thank Reviewer 1 for this comment. Although we did not see any specific mention of the research hypotheses in paragraph 2.1.1, we have revised our language throughout the manuscript to avoid mention of “predictors” and speak to correlations instead.

  1. The title of Table 2 reports “Item Means and Bivariate Correlations Between Predictors and Well-Being Outcomes”. The title is incomplete, as the Table contains also Prosociality Outcomes. Please modify the title.

Response: We thank Reviewer 1 for pointing out that the Table 2 title did not include mention of prosociality, a key construct of interest. We have revised the Table 2 title to “Table 2. Item Means and Bivariate Correlations Between Meaning, Friendships, Engagement, Wellbeing, and Prosociality.”

  1. In paragraph 2.4.4 Authors state: “Finally, we explored whether students’ experience of meaning in the program predicted higher levels of reported prosociality”. Since the analyses are correlational, I suggest not referring to predictive and causal hypotheses.

Response: We agree and have revised the sentence in question to speak to an association. The sentence now read as follows: “Finally, we explored whether students’ experience of meaning in the program was associated with higher levels of reported prosociality.”

  1. In paragraph 2.5 I suggest that more space should be devoted to the discussion of study 1 rather than the introduction to study 2.

Response: We thank Reviewer 1 for this suggestion to devote more space in section 2.5 to discuss Study 1. As requested, we have added a few additional details to the first paragraph in section 2.5 where we summarize the findings of Study 1. We are also open to moving the next two paragraphs introducing Study 2 to section 3 if the editor prefers.   

  1. Study 2. In paragraph 3.2 the SDAge value is incomplete. Please check.

Response: We thank Reviewer 1 for catching this typographical error. Upon reflection, we have fully removed the demographic information initially provided for the sample who visited C2C. Given that several of these children were excluded for not having adequate permission/consent forms, their data should be deleted and not included in our report. Therefore, we provide only the demographics for the final sample that we have ethical approval to analyze (see p. 19).

  1. Discussion and Conclusion. Discussion, Conclusions and Limitations are well argued.

Response: We are happy to see that Reviewer 1 was pleased with the discussion, conclusion and limitations.  

  1. As a limitation, I suggest to refer also to the fact that gender differences were not considered. it would be interesting to explore this further, as the literature reports significant gender differences in both prosociality and well-being.  

Response: We appreciate this suggestion. The revised manuscript now urges future researchers to collect larger samples so that interesting questions regarding gender differences in helping and the emotional rewards of helping can be studied (see p. 25).

Reviewer 2 Report

The authors have presented their nice designs over the teaching prosociality and well-being of children in the real world. The intention of the paper is nice, the data are rich, and the implication is profound. Thus, the paper should be published. Before that the authors might notice the followings:

1. The title might be simply read as “Are real-world prosociality programs associated with greater psychological well-being in primary school-aged children?”

2. The pre- and post-intervention differences in the psychological well-being states might be further analyzed in regards to the age effects, and the possible age trend is meaningful for the related education practice.

3. Boys and girls might behave differently when they imitate the elders or practice their daily prosocial acts, and the authors have the data to clarify these possible effects.

4. Are there any data of extremes, for instance, higher vs lower psychological well-being groups? How would the intervention produce these possible significant effects?

Author Response

  1. The authors have presented their nice designs over the teaching prosociality and well-being of children in the real world. The intention of the paper is nice, the data are rich, and the implication is profound. Thus, the paper should be published. Before that the authors might notice the followings:

Response: We appreciate Reviewer 2’s flattering comments and support of the paper.

  1. The title might be simply read as “Are real-world prosociality programs associated with greater psychological well-being in primary school-aged children?”

Response: We appreciate this suggestion and have shortened the title accordingly.

  1. The pre- and post-intervention differences in the psychological well-being states might be further analyzed in regards to the age effects, and the possible age trend is meaningful for the related education practice.

Response: We thank Reviewer 2 for this suggestion to examine the age trends in Study 2. We conducted exploratory repeated measures ANOVA analyses to examine whether age was associated with or interacted with condition to predict the change from pre- to post-intervention well-being. As reported in a footnote on p. 23 of the revised manuscript, we found no strong evidence that age was significantly associated with or moderated the impact of condition on the pre- to post-intervention difference in well-being (ps > .073; see excerpt pasted below).

“We additionally examined whether age moderated the effect of condition on the average change in happiness from pre- to post-intervention. An exploratory repeated measures ANOVA revealed that age was neither individually associated with nor moderated the effect of condition on the pre- to post-intervention change in well-being (ps > .073).”

  1. Boys and girls might behave differently when they imitate the elders or practice their daily prosocial acts, and the authors have the data to clarify these possible effects.

Response: We appreciate this interesting query from Reviewer 2. To address this question, we conducted exploratory analyses to see if boys and girls reported similar or different levels of engagement in each of the iGen activities, including games/puzzles, art, reading stories, music/dance, and conversations with Elders. As described in a footnote on p. 13 of the revised manuscript (and the excerpt pasted below), we found no gender differences in how often in a typical week boys and girls engaged in art, music/dance, or had conversations with Elders. However, boys reported both playing games/puzzles and reading stories significantly more often than girls. Given our small sample size, we do not have the statistical power to offer any meaningful inferences from these exploratory analyses but report the results for descriptive purposes.

“We conducted exploratory analyses to examine whether the average amount of time students engaged in iGen activities during a typical week would differ between boys and girls. We found no gender differences in how often students engaged in art, music/dance, or had conversations with Elders (ps > .222). Interestingly, compared to girls, boys reported spending more time on average in a typical week playing puzzles/games (Mboys = 2.58, SDboys = 1.38 vs. Mgirls = .92, SDgirls = .669), t(15.9) = 3.78, p = .002, d = 1.60, 95% CI [1.18, 2.01], as well as reading stories (Mboys = 3.33, SDboys = 2.31 vs. Mgirls = 1.08, SDgirls = 1.88), t(22) = 2.62, p = .016, d = 1.12, 95% CI [.31, 1.92].”

  1. Are there any data of extremes, for instance, higher vs lower psychological well-being groups? How would the intervention produce these possible significant effects?

Response: We appreciate Reviewer 2’s question about extreme scores and outliers. As noted in the manuscript, there were only a few outliers detected in Study 1. Given that Study 1 was not an intervention study, it is difficult to know if the iGen program was the cause of these extreme/outlier scores. Indeed, it is possible, for instance, that some students experienced particularly high or low psychological wellbeing or reported exceptionally high levels of engagement with Elders. Importantly, however, we agree that outlier/extreme scores should be noted, which we have done in text. Moreover, findings reported in text do not change substantially when outlier cases are removed.  
